# QUANTIZATION FOR RAPID DEPLOYMENT OF DEEP NEURAL NETWORKS

## ABSTRACT

This paper aims at rapid deployment of the state-of-the-art deep neural networks (DNNs) to energy efficient accelerators without time-consuming fine tuning or the availability of the full datasets. Converting DNNs in full precision to limited precision is essential in taking advantage of the accelerators with reduced memory footprint and computation power. However, such a task is not trivial since it often requires the full training and validation datasets for profiling the network statistics and fine tuning the networks to recover the accuracy lost after quantization. To address these issues, we propose a simple method recognizing channel-level distribution to reduce the quantization-induced accuracy loss and minimize the required image samples for profiling. We evaluated our method on eleven networks trained on the ImageNet classification benchmark and a network trained on the Pascal VOC object detection benchmark. The results prove that the networks can be quantized into 8-bit integer precision without fine tuning.

## 1 INTRODUCTION

Deploying state-of-the-art deep neural networks (**DNN**s) to embedded systems is a challenging task due to the inherent nature of huge number of computations and large memory requirements. These impediments are partly caused by the considerable amount of the redundancies found in the network parameters intended for ease of training. Thus, the parameters encompasses abundant opportunities for trimming strategies, namely pruning and quantizing to low precision (Choi et al., 2017; Courbariaux et al., 2015b; Han et al., 2015). However, running DNN inference on accelerators equipped with fixed-point arithmetic units in an energy efficient manner also requires limiting the precision of the feature maps (Lin & Annapureddy, 2016; Migacz, 2017; Mishra et al., 2017).

Previous works (Courbariaux et al., 2015b; Gysel, 2016; Lin & Annapureddy, 2016; Migacz, 2017) have exhibited converting pretrained DNNs to 8-bit precision does not induce any accuracy loss. The feature maps and the network parameters were quantized at the granularity of layers to accommodate for large diversities in the dynamic range across the layers. Even though they showed good results for a few popular DNNs like AlexNet, VGG-Net, or GoogLeNet, it is not clear whether it would still work for many other recent DNNs with compact architectures.

From our experiments, we were able to observe that this was not the case for some of the recent state-of-the-art DNNs. For example, applying 8-bit quantization to the individual layers of the MobileNet series as done in the previous works showed large accuracy degradations. The excessive accuracy loss could be mitigated by fine tuning the quantized networks (Gysel, 2016; Lin & Annapureddy, 2016). However, in order to reach a competitive level of accuracy for each network with fine tuning, full-size training and validation datasets were needed to be incorporated along with painstakingly long periods of optimization. As many DNN developers only provide the pretrained networks in full precision without the training or the validation datasets from reasons such as privacy or the outright massiveness of the data size, such obstacles hinder rapid and easy deployment of DNNs in full precision to embedded accelerators designed for low precision.

Instead of converting a full precision pretrained network to lower precision suitable for embedded accelerators, it is also possible to train one from scratch by constraining the weights and the activations (Hubara et al., 2016; Mishra et al., 2017; Zhou et al., 2016; Zhuang et al., 2017). However, achieving state-of-the-art accuracy on large benchmarks such as ImageNet classification, increase in the network size in terms of connections (Mishra et al., 2017) or integrating complicated training process (Zhuang

et al., 2017) is required. Nevertheless, these methods have not been proven for various types of network architectures. Considering the fact that GPUs are the most popular devices for training, it is more practical to convert DNNs into lower precision after utilizing the GPUs' full precision data path for training.

In this paper, we introduce a novel technique in which fine tuning is not necessary for 8-bit linear quantization which quantizes the feature maps and the parameters for individual channels instead of layers to accommodate for the inter-channel variations in the dynamic range. We propose manipulating the kernel weights prior to inference for HW-friendly implementation of channel-wise quantization. Our method significantly reduces the accuracy loss caused by quantizing to lower precision without increasing the inference computation cost. The results show that various state-of-the-art DNNs trained on the ImageNet dataset can readily be converted for 8-bit fixed-point accelerators without fine tuning by using a few training samples for profiling.

## 2 LOW PRECISION QUANTIZATION

It is common practice to quantize the activations and the network parameters for each layer to account for the differences in the dynamic range across the layers (Gysel, 2016; Lin & Annapureddy, 2016; Migacz, 2017). Previous implementations such as Ristretto(Gysel, 2016), a fixed-point quantization simulator based on Caffe, reserves three placeholders for the **fractional length**s (defined as the number of required bits for the fractional part of a fixed-point number) per layer, one each for the input and output feature maps (**IFM** and **OFM** respectively) and for the layer parameters (weights and biases). At every layer, IFM, OFM, and the weights are polled separately for max values and the fractional lengths are calculated accordingly. During run-time, the MSBs and LSBs of the parameters and the activations of each layer are clipped to be containable within the given bit-width and the fractional lengths in order to emulate a generic fixed-point H/W implementation. We will use the term **layer-wise quantization** hereafter to describe this scheme in contrast with **channel-wise quantization** proposed in this paper.

A major down-side of the layer-wise quantization is that the inter-channel variations of the feature maps and the weights are not fully accounted for. Since the fractional length is usually selected to cover the maximum value in a layer, the layer-wise quantization tends to cause excessive information loss in channels with a smaller dynamic range. Therefore, accuracy may be degraded significantly and sometimes never recovered even after exhaustive retraining.

### 2.1 CHANNEL-WISE QUANTIZATION

In the channel-wise quantization, the fractional lengths for the feature maps and the weights can be customized for each channel to minimize the impact of low-precision rounding. Each channel of the IFMs and the OFMs has an independent fractional length based on its expected dynamic range while each channel of the kernels has a fractional length which tightly fits its known values.

Figure 1 demonstrates how the IFMs and the kernels from different channels having different fractional lengths in the channel-wise quantization scheme are computed through a convolution layer compared to the layer-wise scheme. In this example, the input and the output of the convolution layer and weights are all bound to 8 bits while the partial sums are allowed to be accumulated in 32 bits as to avoid data loss. The traversal in the layer-wise scheme is straight-forward as there aren't any discrepancies while adding up the partial sums. On the other hand, the channel-wise method must cope with adding partial sums of varying fractional lengths. A naive solution would be to place a shifter in front of the partial sum adder to adjust all the partial sums to have the same fractional length. However, this scheme is not practical since too many shift operations are required. We resolve this complication by pre-coordinating the fractional lengths of the weights.

The fractional length of a partial sum is determined by adding those of the IFM and the kernel being multiplied together. As the partial sums resulting from different input channels will have different fractional lengths, the smallest fractional length across all partial sums is selected as the reference. The red box in Figure 1 depicts this step. Then, the fractional lengths of the kernels in all the other channels are adjusted during the pre-processing stage to produce this reference fractional length when multiplied with their corresponding IFMs. Limitation was set on the amount adjusted so that the minimum value of the modified fractional lengths of the kernels would not be smaller than the

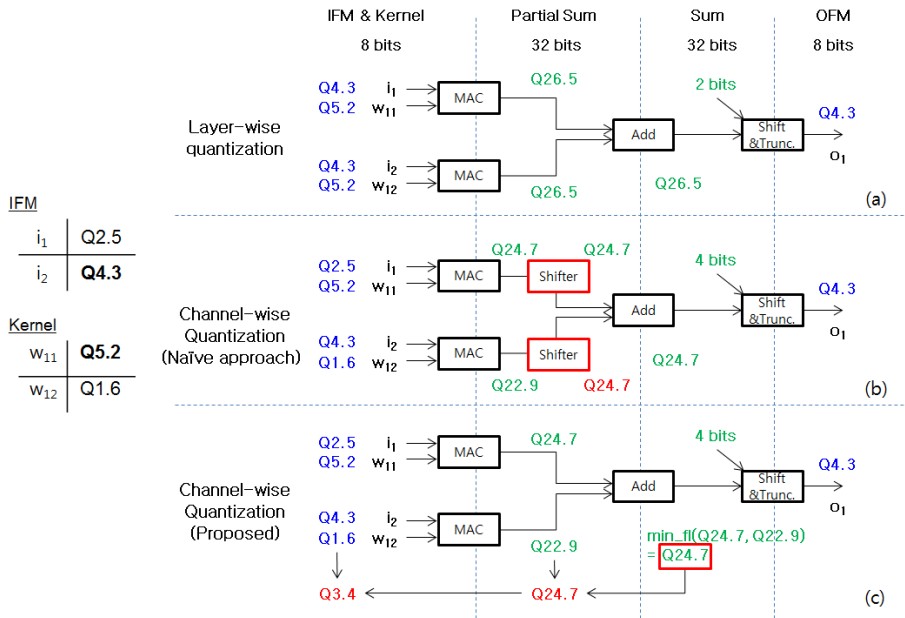

Figure 1: Comparison between layer-wise and channel-wise quantization in a simple convolution layer with 2 IFM channels and 1 OFM channel. $Qn.m$ represents a fixed point integer format with $n$ bits for integer part, $m$ bits for fractional part, and 1 bit for the sign. Total bit-width is equal to $(n + m + 1)$ bits. min_fl(A, B) returns a format with the minimum fractional length. For layer-wise quantization in (a), all the channels of both the inputs and the kernels are forced to share the biggest integer part length across all channels in order to retain the most significant bits. Thus, the number of bits for fractional parts are identical for all paths at the partial sum stage. Channel-wise quantization methods in (b) and (c) could spare more bits for fractional part in accumulator and adder due to channel level granularity, thereby reducing the low precision rounding error during summation. However, the naive approach shown in (b) is not practical since it requires large number of extra bit shifters in front of the partial sum adder to adjust the format of its inputs before adding them. On the contrary, the proposed channel-wise quantization method in (c) does not require such HW cost by considering the bit shifts in the kernel weights in advance.

layer-wise quantization. The overall procedure to determining the channel-wise fractional length is summarized in Algorithm 1.

The channel-wise quantization can be applied to a fully-connected (**FC**) layer by considering each unit as a channel. However, for simplicity, we use the layer-wise quantization for the activations of fully-connected (**FC**) layers. Notwithstanding, the weights of FC layers still needs to be adapted to the preceding layer. Figure 2 shows three such scenarios where we fallback to the layer-wise quantization. In scenario (b) and (d), the activations quantized channel-wise from the previous convolution layer are multiplied with the channel-wise quantized weights of an FC layer which are pre-adjusted to yield the activations having an identical fractional length, hence, the layer-wise quantization for the activations. For scenario (c) where two FCs are stacked consecutively, the layer-wise quantization is utilized throughout the path.

## 2.2 FRACTIONAL LENGTH DETERMINATION

Determining the fractional length (placing the dot in between the integer and fractional part within the given bit-width) is easier said than done. Previous works (Gysel, 2016; Migacz, 2017) profiled the target dataset to look for the max value which became the max representable value in the dynamic range. Profiling, running a network in forward path and collecting statistics, provides either an estimation of the dynamic range when executed during pre-processing stage on a subset of the training dataset or an exact fit when performed during run-time on the actual data being processed.

---

**Algorithm 1** HW-friendly channel-wise quantization.
Profiling dataset is a subset of training data set. $fl$ stands for fractional length ($fl^{ker}$: kernel $fl$, $fl^{ifm}$: input feature map $fl$, $fl^{ofm}$: output feature map $fl$, $fl^{pSum}_{ji}$: $fl$ for partial sum of one kernel and one input, $fl^{adder}_j$: adder $fl$, $fl^{bias}_j$: bias $fl$, $shift_j$: layer output bit-wise shift amount).

---

**Require:** network architecture, network parameters, profiling dataset
**Ensure:** $fl^{ker}$, $fl^{bias}$, $fl^{ifm}$, $fl^{ofm}$, $shift$, quantized network parameters
  1. Profile weights and activations
  Calculate statistics of weights and activations of each channel on profiling dataset
  2. Calculate channel-wise fractional lengths
  For each layer,
      calculate $fl^{ker}$ from statistics for each channel of kernels
      calculate $fl^{ifm}$, $fl^{ofm}$ from statistics of activations for each channel
      $fl^{pSum}_{ji} := fl^{ker}_{ji} + fl^{ifm}_i$ for all $(i,j)$ pairs of input and output channels
      $fl^{adder}_j := \min_i \left( fl^{pSum}_{ji} \right)$ for each $j$
      $fl^{bias}_j := fl^{adder}_j$
      $fl^{ker}_{ji} \leftarrow fl^{ker}_{ji} - \left( fl^{pSum}_{ji} - fl^{bias}_j \right)$
      $shift_j := fl^{bias}_j - fl^{ofm}_j$
  3. Quantize network parameters with $fl^{ker}$, $fl^{bias}$

---

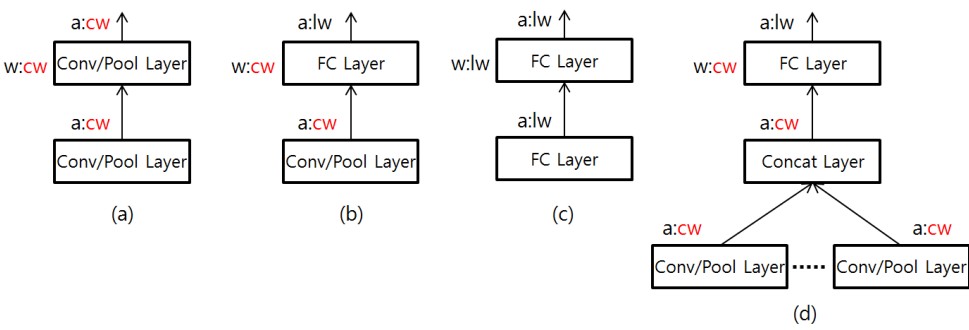

Figure 2: Quantization policy varies with network configurations. $a$ and $w$ represent activation and weights, respectively. ($cw$: channel-wise quantization, $lw$: layer-wise quantization)

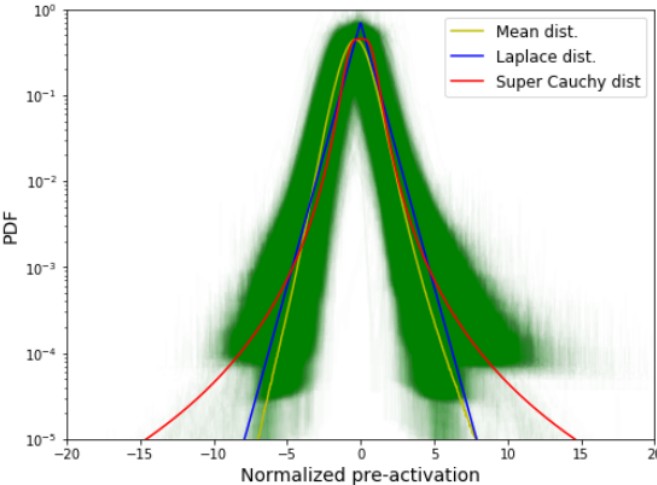

Figure 3: Superpositioned PDFs of pre-activation values of each channel (GoogLeNet w/ ImageNet dataset). Every distribution is normalized to have unit variance. Y-axis is in log scale. Mean dist. represents the averaged PDF of all channels.

The obvious side effect is that selecting just the right size of the dataset to profile is not trivial. A large dataset will increase the chance of electing an outlier as the max value which will overestimate the dynamic range. This will consequently penalize the fractional part of the fixed-point representation. On the other extreme where insufficient size of the profiling dataset is used, the values overflowing the determined fixed-point notation during inference will cause severe performance degradation.

Lin proposed to use the $n$-th moments of the distribution instead of the max value to identify the optimal fixed-point bit-width and the fractional length (Lin & Annapureddy, 2016). The dynamic range of the activation is determined so that the signal-to-quantization-noise-ratio (**SQNR**) caused by quantization would be minimized. In this case, two factors contribute to the quantization noise. The first is the quantization error found within the dynamic range and the latter is the overload error where the values beyond the range are clipped to either the upper or the lower bound. When the number of the quantization levels is fixed, there exists an optimum interval between the levels which makes these two errors balanced. This method is much less sensitive to the size of the profiling dataset since the $n$-th moments of a distribution are more stable measures than the max value. A positive side effect is that optimum interval can be found even with a small dataset as shown in Section 3.1.

Figure 3 illustrates the superpositioned probability density functions (**PDF**s) of the pre-activation values of the individual OFM channels measured on GoogLeNet(Szegedy et al., 2014) trained with the ImageNet dataset. All PDFs were normalized and shifted to have an unit variance and a mean of zero prior to compositing the functions. As can be seen from the graph, there is a large variation in the distribution of the pre-activation values. In Lin & Annapureddy (2016), normal distribution was used to approximate them. However, the mean distribution over all the channels, shown in Figure 3, suggests Laplace distribution rather than normal distribution. We also found that the fractional length obtained by either normal or Laplace distribution tends to underestimate the dynamic range due to the heavy tails of the actual distribution in many channels. In those cases, truncated super Cauchy distribution, defined as follows, provides smaller quantization-induced noise by appropriately considering the tails.

$$f(x) = \begin{cases} \frac{\sqrt{2}}{\pi\gamma\left[1+\left(\frac{x-x_0}{\gamma}\right)^4\right]}, & \text{if} -15 < x - x_0 < 15 \\ 0, & \text{otherwise} \end{cases} \tag{1}$$

Here, $x_0$ is the location parameter and $\gamma$ is the scale parameter.

## 2.3 Exploiting channel-wise PDF

Large variations in distributions across the OFM channels naturally led us to search for the optimal PDF for each channel in determining the fractional length. For this purpose, we constructed a dataset consisting of the best-fit PDFs producing the highest SQNR for the individual OFM channels in GoogLeNet, Inception-v3(Szegedy et al., 2015), and MobileNet(Howard et al., 2017). A simple classifier was trained to select the best-fit PDF during quantization by taking a vector of n-th moments of activation values in each channel. The classifier was trained to choose from truncated super Cauchy or Laplace distribution. We obtained 83% classification accuracy by using the k-nearest neighbors classifier with $k = 12$ (Samworth, 2013).

## 3 Benchmark results

### 3.1 ImageNet classification task

The proposed quantization method was evaluated on various state-of-the-art deep networks trained on the ImageNet dataset containing 1.2M training and 50k validation examples. Pretrained networks were quantized into 8-bit fixed point format by using the profiling dataset sampled from the training set and evaluated on the whole validation dataset (50k examples). Uniform linear quantization was used for all the cases. Batch normalization(Ioffe & Szegedy, 2015) layers were fused into convolution layers before the quantization process. Unsigned integer format was employed for the activation values with the ReLU nonlinearity.

A comparison against the layer-wise quantization is summarized in Table 1. Conventional method with the layer-wise quantization based on the max value provided good quantization results for GoogLeNet, VGG16(Simonyan & Zisserman, 2014), and Inception-v3 which were the most popular networks in the previous quantization and pruning papers. With more recent networks such as MobileNet, MobileNet2(Sandler et al., 2018), ResNet(He et al., 2015), Inception-v4(Szegedy et al., 2016), and Xception(Chollet, 2016), severe accuracy loss was observed. We figured out that the outliers were the major source of accuracy degradation after layer-wise quantization. For example, using the max value of a parameter could significantly overestimate its dynamic range when there are outliers with extraordinarily large values which cannot be seen in the validation set or when deployed. Carefully removing those outliers will significantly improve the quality of quantization even if layer-wise max-based method is used. Thus, there are previous papers showing better results than our baseline layer-wise quantization. However, we did not consider such improvement in the baseline because it requires extra effort and the process itself might taint the dataset since there's no explicitly clear boundary of the outliers.

We evaluated the channel-wise quantization in four modes depending on the method to determine the fractional lengths: MAX, Laplace, S.Cauchy, and PDF-aware. In MAX mode, the max values of the activation tensors were used to decide the factional lengths of the feature maps. As for the Laplace or S.Cauchy modes, the optimal fractional lengths were estimated from the $n$-th moments of the activations by assuming PDF as either Laplace or truncated super Cauchy distribution.

Regardless of the modes, the channel-wise quantization exhibited significantly improved accuracy losses for all the mentioned networks. In the MAX mode, we still observed a large accuracy degradation in the Inception-v4 network. We discovered that there were extremely large outliers in the activations of a few layers causing significant overestimation of their dynamic ranges. This problem can be resolved by applying other modes (Laplace, S.Cauchy, or PDF-aware). The Laplace and S.Cauchy modes showed similar performance overall but different behavior depending on the network. The best result came with the PDF-aware mode which selects the best-fit PDF for each channel.

Figure 4 illustrates the required size of the profiling dataset for the MAX and the OPT methods when measured on Inception-v3. Fractional lengths were calculated based on randomly selected images from the ImageNet training dataset. The MAX method required a large number of samples (¿100) to reach a stable accuracy, whereas a few samples were enough to stabilize the accuracy for the OPT method. Since most of the published networks are trained in full precision while accelerators mandate low-precision representation, being able to readily port a network with just a few training samples is a huge advantage for easy deployment of pretrained full-precision DNNs. Accordingly, the proposed

Table 1: Top-1 accuracy loss after 8-bit quantization in various large scale networks trained on the ImageNet dataset. No retraining is performed. *Reference (Float32)* lists baseline accuracies while all other figures are accuracy losses. Modes for determining the fractional length: *MAX* (reserve integer length to include at least the max value), *Laplace* (optimal fraction length based on Laplace distribution), *S.Cauchy* (optimal fraction length based on truncated super Cauchy distribution), *PDF-aware* (optimal fractional length based on optimum PDF for each channel). Accuracy losses above 1.0% point are in **bold face**.

| Network | Reference (Float32) | Layer-wise | | Channel-wise | | | |
|---|---|---|---|---|---|---|---|
| | | MAX | Laplace | MAX | Laplace | S.Cauchy | PDF-aware |
| GoogLeNet [1] | 68.93% | 0.23% | 0.23% | 0.13% | 0.15% | 0.08% | 0.05% |
| SqueezeNet [2] | 58.39% | **2.02%** | 0.68% | 0.22% | 0.23% | 0.43% | 0.27% |
| MobileNet [3] | 69.50% | **5.48%** | **4.02%** | **1.17%** | 0.66% | 0.66% | 0.73% |
| MobileNet2 [4] | 71.23% | **71.13%** | **71.13%** | **1.73%** | **1.81%** | **3.09%** | **1.68%** |
| VGG16 [5] | 68.34% | 0.29% | 0.19% | -0.01% | -0.04% | 0.01% | -0.06% |
| ResNet101-v2 [6] | 78.04% | **9.52%** | **5.17%** | **1.01%** | 0.74% | **1.58%** | 0.83% |
| ResNeXt50-32x4d [7] | 76.84% | **1.13%** | 0.65% | 0.51% | 0.65% | 0.78% | 0.32% |
| Inception-v3 [8] | 77.97% | 0.99% | 0.66% | 0.23% | 0.09% | 0.18% | 0.24% |
| Inception-v4 [9] | 79.90% | **74.65%** | 0.61% | **26.07%** | -0.06% | 0.07% | 0.13% |
| Incep.-ResNet-v2 [10] | 80.19% | **1.45%** | 0.71% | 0.18% | **1.11%** | 0.64% | 0.36% |
| Xception [11] | 78.72% | **54.72%** | 0.57% | **1.11%** | 0.60% | 0.48% | 0.33% |

[1] Szegedy et al. (2014), [2] Iandola et al. (2016), [3] Howard et al. (2017), [4] Sandler et al. (2018), [5] Simonyan & Zisserman (2014), [6] He et al. (2015), [7] Xie et al. (2016), [8] Szegedy et al. (2015), [9] Szegedy et al. (2016), [10] Szegedy et al. (2016), [11] Chollet (2016)

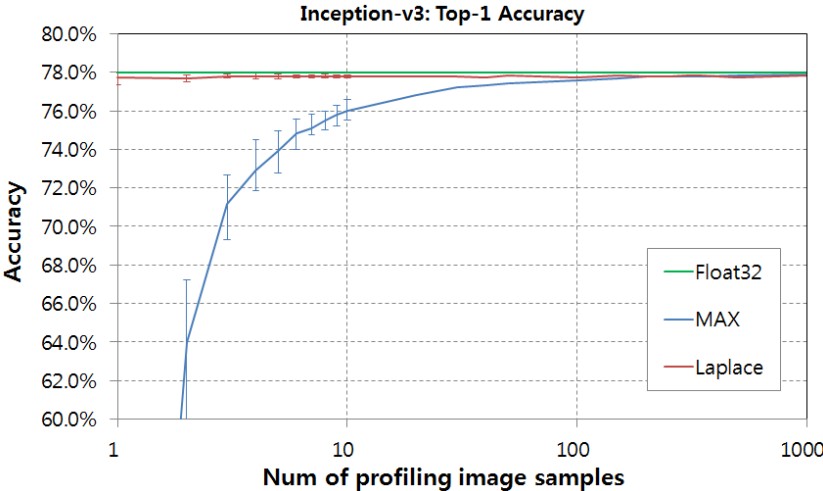

Figure 4: Effect of profiling dataset size on accuracy with quantization for Inception-v3. MAX method requires large number of samples for profiling to reach a stable accuracy. Laplace method stabilizes quickly with a few samples.

quantization method is able to reach a competitive accuracy without the need for profiling a large number of samples or fine tuning.

## 3.2 OBJECT DETECTION

We performed network quantization on YOLO-v2(Redmon & Farhadi, 2016), a state-of-the-art object detection network. The network was trained and tested on the Pascal VOC dataset (Everingham et al., 2015).

Table 2 shows the loss in mean AP after quantization using our method in comparison with the layer-wise quantization. The layer-wise quantization caused 2.5% point drop in mean AP after

Table 2: Loss in mean AP after 8-bit quantization in YOLO-v2 (Redmon & Farhadi, 2016). No retraining performed. 'Reference (Float32)' lists baseline accuracy while all other figures are accuracy losses. Loss above 1.0% point is in **bold face**.

| Network | Reference (Float32) | Layer-wise | | Channel-wise | | | |
|---|---|---|---|---|---|---|---|
| | | MAX | Laplace | MAX | Laplace | S.Cauchy | PDF-aware |
| YOLO-v2 | 72.64% | **2.50%** | **2.25%** | 0.14% | 0.22% | 0.70% | 0.38% |

quantization. However, our method did not suffer from such a problem by selecting the fractional lengths adapted to the individual channels.

## 4 RELATED WORKS

Han quantized the network parameters after pruning for compression in Han et al. (2015). Kim proposed on using Hessian-weighted clustering to achieve a better compression ratio in quantization (Choi et al., 2017). However, in those works, only the network parameters were quantized to save storage space leaving the feature maps in full-precision. Both the activations and the network parameters were quantized layer-wise to accommodate the large variations in the dynamic range across the layers in Courbariaux et al. (2015b); Gysel (2016). The max values found in activations were used to decide on the fractional lengths, and intensive fine tuning were required to recover accuracies degraded by quantization in some networks. Lin used SQNR instead of the max value to minimize the bit-width for each layer and optimized DNNs for fixed-point operation (Lin & Annapureddy, 2016). Migacz achieved linear quantization for 8-bit integer operation without fine tuning by minimizing the information loss with Kullback-Leibler (KL) divergence (Migacz, 2017). Unfortunately, collection of the activation histograms were required from a large number of samples. All in all, these methods used the layer-wise quantization scheme.

Aggressively lowering the precision to be under 4 bits for both the weights and the activations have been actively explored (Courbariaux et al., 2015a; Rastegariy et al., 2016; Hubara et al., 2016; Li et al., 2016; Zhou et al., 2016; Leng et al., 2017; Lin et al., 2017). Although they revealed impressive results on small benchmarks, there is still a huge gap in accuracy on large benchmarks such as the ImageNet classification using state-of-the-art networks trained in full precision. Recent progress shows that it is possible to reduce the precision of DNNs to 4 bits without sacrificing accuracy by increasing the network size (Mishra et al., 2017) or training the networks in multiple stages with guided training (Zhuang et al., 2017). These work focus on training DNNs for low-precision inference from scratch rather than quantizing pretrained full-precision networks.

## 5 CONCLUSION

In this paper, we proposed a set of methods for rapid deployment of DNNs trained in full precision to fixed point accelerators with limited precision computation units. The channel-wise quantization recognizes the inter-channel diversities in the dynamic range of the feature maps. HW cost for implementation is minimized by adjusting the fractional lengths of the kernel parameters. We evaluated our method on eleven state-of-the-art DNNs trained on the ImageNet dataset and an object detection network trained on Pascal VOC dataset. In comparison to the previous method (i.e the layer-wise quantization), the channel-wise quantization can reduce the accuracy loss caused by quantization substantially.

We also showed that quantization requires just a few image samples if we utilize the $n$-th moments of the activations instead of the maximum value. In this way, deployment is possible even when just a few training samples are available for the trained network model. We further improved our method by considering the variations in distribution across the channels. A simple classifier was used for selecting the best-fit PDF for each channel from the statistical features. We were able to accomplish negligible accuracy loss (less than 1% point in eleven networks out of twelve) after quantization without fine tuning.

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
