# OpenReview forum: "Quantization for Rapid Deployment of Deep Neural Networks"
_ICLR.cc/2019/Conference_

### Official Review · AnonReviewer1 · 2018-11-01
**Interesting approach to channel-wise CNN quantization with adaptive bit allocation, with evaluation on eleven modern CNNs, comparison with simple layer-wise baseline**

**Rating:** 5
**Confidence:** 4

**Review:**

This paper proposes a technique for channel-wise quantization of CNNs
to 8-bit, fixed point precision. The authors propose several
techniques for analyzing the statistical properties of output channel
activations in order to select the best fractional bit length for each
channel. Experimental results on eleven different CNN architectures
demonstrate that the approaches proposed result in significantly less
accuracy loss when compared to a layer-wise baseline.

The paper has the following strengths:

 1. The experimental results on eleven different architectures (of
    varying depth and breadth) are convincing, and are consistently
    better than layer-wise MAX for choosing fractional bit length.

The paper has the following weak points:

 1. There is not much coherence between the description of the
    approach in section 2.1, Figure 1, Algorithm 1, and
    Figure 2. Notation is used in Algorithm 1 which is never defined.
 2. Related to the previous point, the proposed technique has a lot of
    moving parts and I don't feel that it would be easy to reproduce
    the results of the paper. There are some vague statements, like
    "We resolve this complication by pre-coordinating the fractional
    lengths of the weights", which require significantly more
    precision. This issue -- one of the main issues with channel-wise
    versus layer-wise quantization -- is never returned to in the
    definition of the method.
 3. The experimental comparison with layer-wise quantization is
    somewhat lacking. Is layer-wise MAX the state-of-the-art in CNN
    quantization? The results comparing channel-wise and layer-wise
    MAX are already convincing, but are the moment-analysis approaches
    not equally applicable to layer-wise quantization?
    State-of-the-art results that are less sensitive to outliers
    should be included in Table 1. A comparison with layer-wise
    approaches would be nice to have also in Figure 4 to show
    sensitivity to profiling set size.

The experimental results in the paper are impressive, and the analysis
motivating the approach is convincing. However, there are presentation
and clarity issues in the technical development, and the comparative
analysis is lacking broader comparisons with the state-of-the-art (to
be fair, the authors recognize that layer-wise MAX as a baseline is
particularly susceptible to outliers). These two aspects combined,
however, lead me to the opinion that this work is just not quite ready
for publication at ICLR.

---

> ### Author Response · Authors · 2018-11-19
> **Response to Reviewer**
>
> Thank you very much for your helpful comments and suggestions.
> 1. Notations have been added to Algorithm 1 and modified Figure 1 along with its description, as suggested.
> 2. The channel-wise quantization algorithm is calculated statically as show in Algorithm 1 and exemplified in Figure 1. Algorithm 1 demonstrates all aspects of “pre-coordinating the fractional lengths of the weights.” Therefore, should be able to be reproduced by anyone and didn’t think it would be necessary to write it out in words.
> 3. The layer-wise moment-analysis results have been added to the paper in Table 1 and 2. Channel-wise out-performs layer-wise in most cases without the need for additional HW. The overhead introduced in channel-wise quantization is just a simple pre-processing step of determining the fraction lengths by executing Algorithm 1.
> We would like to emphasize that our paper proposes a practical solution for channel-wise quantization. Naïve channel-wise quantization requires adding huge number of HW shifters and providing values for them which make it unrealistic for implementation (please see Figure 1 in the revised manuscript). The proposed method does not require such cost by manipulating the kernels prior to inference. For your reference, Figure 1 has been modified to make the distinction clearer.

---

### Official Review · AnonReviewer2 · 2018-11-05
**Novelty is limited**

**Rating:** 5
**Confidence:** 4

**Review:**

This paper proposes an new 8-bit quantization strategy for rapid deployment.

8-bit quantization has attracted many attentions recently. And it is already well used in GPU servers (cudnn), phones, ARM chips and various ASIC neural network chips. In these situations, almost no performance drop is observed for classification and detection tasks.

So, the novelty of this paper is limited.

---

> ### Author Response · Authors · 2018-11-19
> **Response to Reviewer**
>
> Thank you very much for your comments. Competing methods in other papers require retraining or needs to cope with high accuracy loss when quantized in a layer-wise fashion. The proposed method is the first of its kind to resolve these issues by incorporating channel-wise quantization and moment-analysis method which DOES NOT require retraining or the training dataset. Naïve channel-wise quantization requires adding huge number of HW shifters and providing values for them which make it unrealistic for implementation (please see Figure 1 (b) in the revised manuscript). The biggest contribution of our paper is the HW-friendly channel-wise quantization by manipulating the kernels prior to inference. For your reference, Figure 1 has been modified to make the distinction clearer.

---

### Official Review · AnonReviewer4 · 2018-11-13
**promising method for 8-bit quantization without sensitivity to outliers, limited in novelty and presentation clarity**

**Rating:** 5
**Confidence:** 3

**Review:**

The paper proposes channel-wise 8-bit quantization rather than layer-wise. It further takes advantage of work using moment analysis instead of just MAX values to avoid susceptibility to outliers. The main take-away seems to be that channel-wise set ups limit the need for outlier removal and the care with which you select your data subset when performing quantization.

Pros:
- using channel-wise quantization (with MAX values or moment-analysis) yields improvement over layer-wise MAX approaches
- limits the amount of care that is needed to be taken when applying quantization (e.g. size of data subset used)
- shows differences in degradation when blindly applying quantization methods to different networks; with less (but still some) variation in degradation when applying channel-wise quantization

Cons:
- unclear how much is gained over layer-wise and MAX value methods with careful tuning/removal of outliers; would be good to see if careful tuning closes the gap or if channel-wise methods are the clear winner
- unclear if the layer-wise set up with moment-analysis could help to avoid the need for outlier removal altogether and (potentially) offer similar improvements to the channel-wise set up; a few more experiments are important to determine specifically if improvement is with respect to channel-wise or moment-analysis since only layer-wise MAX results are presented
- clarity, presentation, and organization can be improved to help with flow, avoid confusion, and improve readability

Overall:
The paper offers nice empirical results regarding the relative ease with which one can quantize networks when considering channel-wise quantization (and moment-analysis), but the overall novelty is limited. With the limited novelty, the primary benefits appear to be the ease of quantization for rapid deployment and channel-wise setups. Comparisons with stronger baseline numbers when using layer-wise methods would give a more complete picture. In addition, having these stronger tuned baseline numbers on even more networks would be great to show that the channel-wise method has clear impact across the board, even with respect to well-tuned layer-wise baselines. These results could give better support for the importance of the novelty.

---

> ### Author Response · Authors · 2018-11-19
> **Response to Reviewer**
>
> Thank you very much for your helpful comments. We have addressed them as follows:
>
> 1. Results of layer-wise quantization with moment-analysis method were added
> : The layer-wise quantization with moment-analysis method could remove the effect of outliers. To show this clearly, we added the results of applying moment-analysis method to layer-wise quantization in Table 1 and 2 in the revised manuscript. However, even if outliers were to be removed, channel-wise method consistently outperforms layer-wise one. Channel-wise method in comparison to the layer-wise method both with moment-analysis (or MAX) clearly demonstrates that channel-wise has the advantage.
>
> 2. Novelty of the paper
> : The main contribution of the paper is manipulation of the weights prior to inference for channel-wise quantization as shown in Figure 1 and Algorithm 1. Naïve channel-wise implementation requires shifters to be in place for each channel which would cause huge hardware cost (please see Figure 1 (b) in the revised manuscript). Thus, it is not a practical solution at all in HW perspective. The proposed method performs channel-wise quantization WITHOUT the need for any additional HW and we believe there’s novelty in that it enables channel-wise quantization for real-world HW. We modified Figure 1 and improved manuscript to reflect this point clearly.

---

> > ### Comment · AnonReviewer4 · 2018-11-26
> > **Thank you for the updates**
> >
> > Thank you for the updates. These results indicate more consistent results for the non-MAX approaches for layer-wise methods. Overall, channel-wise methods do outperform layer-wise, but the gap is smaller now. The take-away is that channel-wise methods (with Laplace, S. Cauchy, or PDF-aware modes) will reliably improve classification accuracy loss over layer-wise methods (as well as AP in object detection) since large (>4%) degradation is not present in channel-wise settings.
> >
> > Overall, I think this is a nice result, but it is lacking in presentation and analysis. The overall presentation can be cleaned up and improved with respect to the plots and graphics to help better guide the reader through the paper.
> >
> > For example:
> > - Further analysis in the way of plots or tables to compare relevant network statistics (to better contextualize those which exhibited such awful degradation in the layer-wise/channel-wise MAX cases)
> > - The algorithm and figure on page 4 are made less effective by being isolated from the flow of the paper. It is understandable that all the text related to a figure may not be on the same page, but in this case there is opportunity to improve the formatting to allow better flow and readability.
> > - The plot on pages 5 would be more powerful if it showed statistics from other networks as well to give a better picture of the variability between the networks
> > - The plot on page 7 could be more informative if it showed both layer-wise and channel-wise cases as well as depicted more networks
> > - Minor: including top-5 accuracies as well (since differences in top-1 accuracy degradation are often <0.5%)
> >
> > Other things that I think would be interesting but understandably hard to add would be exploration of other tasks (as done with object detection/YOLO system), even more networks, and including some results from systems that use methods with retraining such that those numbers can be seen for reference when making the trade-off.

---

> > > ### Author Response · Authors · 2018-12-02
> > > **Thank you very much for the valuable comments.**
> > >
> > > Thank you very much for the valuable comments. We will improve the manuscript in the final version by adding further analysis and results as you suggested.

---

### Meta-Review · Area_Chair1 · 2018-12-17
**lacks novelty**

**Confidence:** 5
**Recommendation:** Reject

**Metareview:**

This paper proposes an 8-bit quantization strategy for rapid DNN deployment. 3 reviewers all rated this paper as marginally below acceptance threshold due to lack of novelty. 8 bit quantization (including channel-wise) is a well studied task. The paper lacks comparison with peer work.